# Transcriptional Stress Memory and Transgenerational Inheritance of Drought Tolerance in Plants

**DOI:** 10.3390/ijms232112918

**Published:** 2022-10-26

**Authors:** Nguyen Hoai Nguyen, Nam Tuan Vu, Jong-Joo Cheong

**Affiliations:** 1Faculty of Biotechnology, Ho Chi Minh City Open University, Ho Chi Minh City 700000, Vietnam; 2Center for Food and Bioconvergence, Seoul National University, Seoul 08826, Korea

**Keywords:** drought tolerance, chromatin remodeling, chromatin loop, non-coding RNA, stress memory, transgenerational inheritance

## Abstract

Plants respond to drought stress by producing abscisic acid, a chemical messenger that regulates gene expression and thereby expedites various physiological and cellular processes including the stomatal operation to mitigate stress and promote tolerance. To trigger or suppress gene transcription under drought stress conditions, the surrounding chromatin architecture must be converted between a repressive and active state by epigenetic remodeling, which is achieved by the dynamic interplay among DNA methylation, histone modifications, loop formation, and non-coding RNA generation. Plants can memorize chromatin status under drought conditions to enable them to deal with recurrent stress. Furthermore, drought tolerance acquired during plant growth can be transmitted to the next generation. The epigenetically modified chromatin architectures of memory genes under stressful conditions can be transmitted to newly developed cells by mitotic cell division, and to germline cells of offspring by overcoming the restraints on meiosis. In mammalian cells, the acquired memory state is completely erased and reset during meiosis. The mechanism by which plant cells overcome this resetting during meiosis to transmit memory is unclear. In this article, we review recent findings on the mechanism underlying transcriptional stress memory and the transgenerational inheritance of drought tolerance in plants.

## 1. Introduction

The global climate crisis is reducing rainfall, resulting in long periods of dry weather and repeated droughts that seriously threaten crop productivity and the food supply. Plants are unable to escape from adverse environments and thus deal with stressful conditions by regulating the expression of tolerance genes to induce physiological and cellular responses [1,2,3]. Under drought conditions, plants upregulate the level of endogenous abscisic acid (ABA), which is a major plant hormone in the cellular tolerance response to osmotic stress [4]. ABA induces the closure of stomatal apertures on the leaf epidermis, to limit transpiration and thereby prevent the loss of water [5]. In addition, ABA induces the expression of numerous genes encoding enzymes that catalyze the biosynthesis of osmoprotectants, which mitigate stress and promote plant tolerance [6,7].

To initiate gene transcription, the surrounding chromatin must be converted from a repressive to an active state to enable access by transcriptional activators and RNA polymerases [8]. In the chromatin of eukaryotic cells, genomic DNA is compacted into the nucleus by wrapping around histone octamers. Chromatin architecture is epigenetically altered by remodeling through DNA (de)methylation, alterations in nucleosome density and composition, and histone modification, which take place at the promoter, the transcription start site (TSS), and gene-body regions [9,10]. In addition, in response to environmental signals or developmental cues, non-coding RNAs (ncRNAs) are generated from intergenic regions, repetitive sequences, transposons (TEs), and pseudogenes, and interact with their targets to inhibit gene expression at the transcriptional, posttranscriptional, and epigenetic levels by promoting mRNA cleavage or repressing translation [11].

Plants pre-exposed to stress often grow better under subsequent stressful conditions [12,13,14]. This phenomenon has been referred to as stress memory, priming, training, acclimation, and imprinting [15,16,17]. During and after stress, defense signaling metabolites and transcription factors accumulate in plant tissues and may play a role in transient or short-term memory. However, the most plausible mechanism involves epigenetic changes in the chromatin architecture of certain stress-responsive genes called ‘stress memory genes’ that are expressed at highly elevated or reduced levels in response to repeated stress [18,19,20,21]. Lämke and Bäurle [21] defined the sustained differential responses in gene expression (activation or repression) after an exogenous cue with the term ‘transcriptional memory’. An epigenetically modified status acquired during stress can be transmitted to newly developed cells during mitotic cell division [22,23]. Thus, the chromatin architecture constituted by epigenetic marks under initially stressful conditions may pass through the cell division process without alteration, thereby providing mitotic memory [24,25].

Stress memory whose duration is limited to one generation of organisms was termed as ‘somatic stress memory’ [21]. Furthermore, plants can transmit traits acquired during growth to their progenies (transgenerational inheritance) [21,26,27]. In mammalian cells, the acquired memory is completely erased and reset during meiosis. The mechanism by which plant cells overcome this resetting during meiosis and transmit the stress memory to progenies is unclear.

The memory and transgenerational inheritance of stress tolerance have been explored in the context of crop breeding and yield stability [28,29,30]. However, the adoption of tolerance priming (e.g., pre-treatment with stressful conditions) and inheritance technologies has been hampered by an insufficient understanding of their principles. Understanding the molecular mechanisms underlying stress memory and transgenerational inheritance would facilitate the breeding of crops to better be able to withstand adverse climatic conditions.

Recently, Yao et al. [31] summarized recent progress in the regulation of drought response by various transcription factor families in plants. Halder et al. [32] reviewed recent studies on chromatin restructuring under abiotic stresses, crosstalk between epigenetic regulators, and stress priming in plants. In addition, Sun et al. [33] summarized and discussed mechanisms involved in plant stress memory and genome editing technologies with their future possible applications in the breeding of crops for abiotic stress tolerance. Here, we review the research on the epigenetic molecular mechanisms underlying transcriptional drought stress memory in plants. In particular, we infer a hypothetical concept for the possible mechanisms underlying the epigenetic regulation of transcriptional and the transgenerational transmission of drought stress tolerance, using associations with observations of plant reproductive process and reprogramming events in germ cells of other animal systems.

## 2. Transcription of Drought-Responsive Genes

### 2.1. Drought-Responsive Genes

More than half of the genes up- or downregulated under drought or high-salinity conditions are also regulated by ABA application, suggesting that the expression of osmotic stress-responsive genes is driven mainly by ABA [6,7,34]. ABA induces the expression of numerous genes encoding enzymes that catalyze the biosynthesis of osmoprotectants such as trehalose and late embryogenesis-abundant proteins. In addition, ABA-induced genes include those encoding proteins exerting either positive or negative effects on its accumulation (biosynthesis, catabolism, and glucose-conjugation), transportation, and signaling network [35]. By analyzing the promoters of ABA-responsive genes, a conserved *cis*-acting ABA-responsive element (ABRE; PyACGTGG/TC) was identified [36]. Subsequently, several ABRE-binding (AREB) proteins and ABRE-binding factors (ABFs) were identified by yeast one-hybrid screening [37,38].

However, a few drought-inducible genes do not respond to ABA, implying that ABA-independent pathways regulate the drought response [39]. The promoters of these genes contain the *cis*-acting element DRE (dehydration-responsive element)/CRT (C-repeat), which functions in ABA-independent gene expression [36]. The ERF/AP2 family transcription factors CBF/DREB1 (C-repeat-binding factor/DRE binding 1) and DREB2, which bind to DRE/CRT elements, were identified in plants [36]. Most CBF/DREB1 target genes in Arabidopsis (*Arabidopsis thaliana*) contain the DRE motif with a conserved (A/G)CCGACNT sequence in their promoter regions. Rice (*Oryza sativa*) genome sequence analyses identified 10 *OsDREB1*s and 4 *OsDREB2*s, indicating that similar transcription factors function in drought stress tolerance in dicotyledonous and monocotyledonous plants. The Arabidopsis *RD29A* (*responsive to desiccation 29A*) contains both *cis*-acting elements, ABRE and DRE/CRT, in the promoter region.

### 2.2. Regulation of ABA Signaling

The ABA de novo synthesis pathway has been unraveled [4,40]. ABA is biosynthesized in vascular tissues and transported to targets including guard cells via the export and import of transporters. In Arabidopsis, three transporter families have been identified: ATP binding cassette (ABC), detoxification efflux carriers (DTX)/multidrug and toxic compound extrusion (MATE) proteins, and nitrate transporter 1/peptide transporter family (NPF) [35,41,42,43]. For instance, ABCG25 exports ABA to the apoplastic space in vascular tissues, while ABCG40 and NPF4.6 import ABA into guard cells in response to osmotic stress.

In target cells, ABA is perceived by soluble receptors in the nucleus and cytosol. Several synonymous ABA receptors, i.e., PYR (pyrabactin resistance), PYL (PYR-related), and RCAR (regulatory component of the ABA receptor), have been identified in Arabidopsis [44,45]. In the absence of ABA under non-stressful conditions, clade A type 2C protein phosphatases (PP2Cs) in guard cells counteract a family of protein kinases, known as sucrose non-fermenting 1-related protein kinase 2s (SnRK2s) via physical interactions, thus providing negative feedback regulation of ABA signaling [46,47] (Figure 1A). Under osmotic stress, following ABA perception, PP2Cs bind to ABA receptors to capture ABA and form the PYL-ABA-PP2C complex; SnRK2s subsequently dissociate from inactivated PP2Cs to restore kinase activity [48] (Figure 1B).

Activated SnRK2s in the cytosol facilitate the functioning of slow anion channel 1 (SLAC1), K^+^ channel 1 (KAT1), and NADPH oxidases in the guard cell membrane to induce stomatal closure [35,49]. In the nucleus, under osmotic stress, activated SnRK2s phosphorylate and activate a family of basic-domain leucine zipper (bZIP) transcription factors, the AREB/ABFs, thereby inducing the expression of numerous ABA-responsive genes [50]. Among the nine AREB/ABFs in Arabidopsis, ABF1, AREB1/ABF2, AREB2/ABF4, and ABF3 act as master transcription factors in ABA signaling to promote osmotic stress tolerance [51].

## 3. Chromatin Modifications for Drought-Responsive Gene Transcription

### 3.1. Drought Stress Tolerance and DNA Methylation

In the nuclear genomes of higher plants and animals, some cytosine bases in CG, CHG, and CHH sites (where H indicates A, T, or C) are methylated [52]; 5-Methylcytosine (5mC) is the most common, but small amounts of N4-methylcytosine and N6-methylcytosine are also present in genomic DNA. Zilberman et al. [53] mapped DNA methylation in the entire Arabidopsis genome at high resolution and revealed DNA methylation in large portions of genes and TEs. In plants, de novo DNA methylation is mediated by the RNA-directed DNA methylation (RdDM) pathway [54]. To initiate the RdDM, 24-nucleotide (nt) small interfering RNAs (siRNAs) are produced by RNA polymerase IV (Pol IV) and direct DNA methylation to homologous Pol V-transcribed loci, thereby inducing transcriptional silencing at repetitive DNA sequences including all types of TEs. The maintenance of methylation and active demethylation in the promoter and gene body regions is mediated by various methyltransferases and by the 5mC DNA glycosylase DEMETER (DME), respectively [55,56].

DNA methylation is a conserved epigenetic mark important for maintaining genome stability and regulating gene transcription [57]. High-resolution, genome-wide analyses suggest that the methylation of gene promoter regions suppresses the transcription of the corresponding gene, whereas the methylation of the gene body may enhance transcription [58,59,60]. Zhang et al. [58] reported that more than one-third of genes expressed in the Arabidopsis genome contained methylation in their transcribed regions, compared to ~5% in their promoter regions. Genes methylated in the transcribed regions were highly expressed, whereas promoter-methylated genes showed tissue-specific expression. In the rice genome, Li et al. [59] reported the repression of promoter-methylated genes, whereas gene-body methylation was associated with gene expression. Additionally, methylation in gene transcriptional termination regions significantly repressed gene expression, to a greater degree than promoter methylation. Xu et al. [60] analyzed changes in gene expression in apple (*Malus domesticus*) under water deficit. Promoter-methylated genes showed lower expression levels than promoter-unmethylated genes. They also found that the hypomethylated status of TEs was due to the demethylation of CHH. Van Dooren et al. [61] reported that mild drought induced changes in DNA methylation in Arabidopsis; this was restricted to CHH sites and predominantly affected TE sequences. The gene promoter methylation-induced repression of transcription is implicated in the silencing of endogenous genes and TEs [62].

There is a link between drought stress tolerance and DNA methylation in plants. For instance, in rice plants, drought-susceptible genotypes predominantly exhibited hypermethylation under drought, whereas drought-tolerant genotypes showed hypomethylation [63,64,65,66]. Zheng et al. [66] observed numerous differentially methylated regions (DMRs) in all 12 rice chromosomes, mainly in gene promoter and exon regions. The tolerant genotypes exhibit not only hypomethylation but also a higher ability for the flexibility of DNA methylation levels. Wibowo et al. [67] found that distinct regions of the Arabidopsis genome are susceptible to DNA (de)methylation in response to hyperosmotic stress. In the tomato (*Solanum lycopersicum*) genome, González et al. [68] showed that brief exposure to simulated drought conditions increased methylation in the coding region, but decreased methylation in the regulatory region, of the drought-responsive gene *SL**Asr2* (*ABA stress ripening 2*). In black cottonwood (*Populus trichocarpa*), drought significantly increased 5mC levels in upstream (2 kb), downstream (2 kb), and repetitive sequences [69]. Chwialkowska et al. [70] reported that the overall DNA methylation level in the barley (*Hordeum vulgare*) genome was high under water-deficiency conditions but recovered to the basal level upon rewatering. Lu et al. [71] detected drought-induced hypermethylation in the upland cotton (*Gossypium hirsutum*) genome, which was restored almost to the pre-treatment level after rewatering.

Under drought stress, the genome-methylation level is altered by the differential regulation of 5mC METs and DMEs in many plant species [57,72]. Kim et al. [73] showed that active DNA demethylation in Arabidopsis depends on the activity of ROS1 (repressor of silencing 1), which directly excises 5mC from methylated DNA. *ros1* mutants exhibited decreased expression of several ABA-inducible genes, in which >60% of the proximal regions became hypermethylated, indicating that a subset of ABA-inducible genes is regulated by ROS1-dependent DNA demethylation.

### 3.2. Chromatin Remodeling Implicated in Drought-Responsive Gene Expression

#### 3.2.1. Histone Modifications

In eukaryotic cells, genomic DNA is packed into the nucleus by wrapping around histone (H) octamers consisting of H2A, H2B, H3, and H4, thereby forming nucleosomes. To alter the activity of a gene, the *N*-terminal side of basic residues such as lysine and arginine in histone tails are covalently modified by acetylation, methylation, phosphorylation, and ubiquitination. Histone modification can have different effects depending on the type of modification and which residue is modified. It is difficult to establish a universal rule for the histone modification effects because the meaning of histone modifications in relation to gene transcription is dependent on the species.

Typically, to suppress gene transcription, certain lysine (K) residues in core histones are deacetylated by histone deacetylases (HDAs) to generate a tight DNA-histone interaction, which hinders access by RNA polymerases. Mehdi et al. [74] reported that a histone deacetylation complex containing HDA19 repressed the expression of ABA receptor genes, such as *PYL4*, *PYL5*, and *PYL6*. Chen et al. [75] showed that HDA6 modulated the expression of the Arabidopsis abiotic stress-induced genes *ABI1* (*ABA-insensitive 1*), *ABI2*, *DREB2A*, *RD29A*, and *RD29B*. Luo et al. [76] reported that the Arabidopsis histone deacetylase 2C (HD2C) interacts physically with HDA6 to suppress the expression of PP2C genes, including *ABI1* and *ABI2*. A number of studies suggest that repressive transcription factors (repressors) bind to the promoter region and recruit corepressors associated with HDAs to induce tight nucleosome–DNA binding (Figure 2A). Ryu et al. [77] showed that the transcriptional repressor BES1 (brassinosteroid insensitive1-ethylmethane sulfonate suppressor) forms a complex with HDA19 and the TOPLESS (TPL) corepressor via the EAR (ethylene-responsive element binding factor-associated amphiphilic repression) motif (LxLxL). The BES1-TPL-HDA19 complex suppresses the transcription of *ABI3*, and consequently *ABI5*, which are major ABA signaling transcription factor genes. Nguyen and Cheong [78] showed that the Arabidopsis EAR motif-containing transcriptional repressor AtMYB44 binds to the promoters of PP2C genes (*ABI1*, *ABI2*, and *HAI1*) and forms a complex with TPL-related (TPR) corepressors. AtMYB44 is a transcription factor that mediates drought and salt stress tolerance in Arabidopsis [79]. Kahn et al. [80] showed that the Arabidopsis protein POWERDRESS (PWR) forms a complex with HDA9 to interact with the transcription factor ABI4, which epigenetically regulates ABA-dependent drought stress tolerance by modifying the acetylation of *CYP707A1* (which encodes a major enzyme of ABA catabolism).

Conversely, to initiate the transcription of a gene, the surrounding chromatin needs to be remodeled to enable access by transcriptional activators and RNA polymerases (Figure 2B). Therefore, gene transcription must be accompanied by nucleosome disassembly or replacement with other nucleosomes of different compositions [81,82,83]. In parallel, histone acetyltransferases (HATs) acetylate histones, thereby neutralizing the positive charges of histone tails to reduce their affinity for negatively charged DNA; this results in weak DNA–histone interactions. Thus, histone acetylation, as with the H3 acetylation of lysine 9 (H3K9ac) and lysine 27 (H3K27ac), is strongly associated with the activation of gene transcription [19,84].

In addition to histone acetylation, histone methylation, which is mediated by histone methyltransferases (HMTs), as with H3 lysine 4 trimethylation (H3K4me3) and H3 lysine 27 trimethylation (H3K27me3), is closely associated with the regulation of gene expression [19,84]. As an example, for the transcription of PP2C genes (e.g., *ABI1*, *ABI2*, and *HAI1*), AtMYB44 repressors are released, and nucleosomes are dissociated from promoter regions in response to salt stress [85]. Under these conditions, histone H3 acetylation (H3ac) and H3K4 methylation (H3K4me3) around TSS regions significantly increased. Wang et al. [86] demonstrated that transcriptional activators (ABFs) bind to PP2C gene promoters to induce gene transcription in response to salt stress. The salt-induced increases in PP2C gene transcription were reduced in *abf3* plants [85], indicating that ABF3 activates PP2C gene transcription.

The chromatin architecture around the promoter and TSS is an important element of the plant response to drought stress [19,87,88]. Li et al. [89] summarized recent findings on histone acetylation changes in these regions during the plant response to drought stress. Kim et al. [90] observed that the nucleosome occupancy of the promoter regions of Arabidopsis drought-responsive genes, including *RD20*, *RD29A*, *RD29B*, and *RAP2.4* (*related to AP2.4*), is low and gradually decreases in response to drought stress. H3K4me3, H3K9ac, H3K23ac, and H3K27ac were enriched on the coding regions of *RD20*, *RD29B*, and *RAP2.4*, which were correlated with gene activation. Ding et al. [91] observed that dehydration stress increased H3K4me3 deposition on the NCED3 (9-*cis*-epoxycarotenoid dioxygenase) gene, which encodes the rate-limiting enzyme of ABA biosynthesis.

In contrast to H3K4me3, which is an active epigenetic mark, H3K27me3 and H3K9me2 are gene silencing marks related to a chromatin repression mechanism for gene expression. In a study of the response to drought stress of a moss (*Physcomitrella patens*) genome, Widiez et al. [92] observed synergistic relationships for the activating marks H3K4me3, H3K27ac, and H3K9ac, and an antagonistic relationship between the H3K27me3 and H3K27ac. Chen et al. [75] reported that ABA induced H3K9K14ac and H3K4me3 but decreased H3K9me2 in several ABA- and abiotic stress-responsive genes. Mao et al. [93] showed that MITE (miniature inverted-repeat transposable element) inserted in the promoter of *ZmNAC111* repressed the gene expression via RdDM and H3K9me2 deposition. ZmNAC111 is a maize (*Zea mays*) NAC transcription factor gene that improves water use efficiency by upregulating drought-responsive genes under water stress. Wu et al. [94] showed that the histone demethylase JMJ30 (JUMONJI-C domain-containing protein 30) removed H3K27me3 from the *SnRK2.8* promoter region, thereby activating its expression and that of the ABA-dependent transcription factor ABI3.

#### 3.2.2. Chromatin Remodeling Complexes

Chromatin remodeling complexes regulate gene transcription by stabilizing or removing nucleosomes occupying the binding sites for transcription factors [95,96]. The SWI/SNF (switch/sucrose non-fermenting) complex is a well-characterized chromatin-remodeling complex in plants. The SWI/SNF complex subunit BRAHMA (BRM) hydrolyzes ATP to supply the energy necessary to modulate the interaction of nucleosomes with DNA, thereby altering their position and occupancy status [97,98]. BRM contains a conserved glutamine-leucine-glutamine (QLQ)-rich motif that mediates protein–protein interactions, an ATP-binding domain implicated in the transcriptional activation of downstream genes, and a carboxy-terminal containing three DNA-binding regions [99]. Among the putative DNA-binding domains, an AT-hook motif allows non-specific DNA binding, and a bromo-domain interacts with acetylated lysine residues of histones H3 and H4 [100].

Whole-genome mapping and transcriptome analyses have revealed that the BRM complex occupies thousands of sites in the Arabidopsis genome, where it contributes to the activation or repression of gene transcription [101]. However, it is unclear how BRM-containing SWI/SNF complexes access and occupy their target loci, even though Arabidopsis BRM contains several DNA- and nucleosome-binding regions. In a study of vegetative development and flowering, the BRM complex was recruited to specific loci via physical interaction with a plant-specific H3K27 demethylase (REF6) that targets certain genomic loci [102]. In a yeast-two-hybrid interactome study, Efroni et al. [103] identified 210 unique transcription factors belonging to 25 families that recruit SWI/SNF complexes to the genomic loci that they regulate. However, the relevance of these interactions *in planta* has not been investigated.

BRM is an important component of the ABA signaling pathway in Arabidopsis. Han et al. [104] observed hypersensitivity to ABA and increased drought tolerance in sprouting Arabidopsis *brm* mutants. In the absence of drought stress, at a transcription factor gene locus (*ABI5*), the BRM complex represses ABA responses by destabilizing the associated nucleosome, thus suppressing gene transcription. Peirats-Llobet et al. [105] reported that the C-terminal region of BRM is phosphorylated by SnRK2.6/OST1 and dephosphorylated by PP2CA. The SnRK2-mediated BRM phosphorylation regulates the BRM-mediated inhibition of *ABI5* expression. By contrast, PP2CA-mediated dephosphorylation restored the ability of BRM to repress the ABA response. Thus, it was proposed that a phosphorylation-based switch mediated by SnRK2s and PP2Cs, in addition to their mutual inhibition via physical interaction, controls BRM-associated chromatin remodeling and regulates the transcription of ABA-responsive genes. Nguyen et al. [85] observed markedly enhanced PP2C gene expression in *brm-3* plants in response to salt stress. Yang et al. [106] examined the three-dimensional chromatin structure of the ATPase subunit of Arabidopsis remodeling complexes, including SWI/SNF, and found that these complexes regulate the linear nucleosome distribution pattern and density to promote H3K27me3 deposition, thereby modulating the chromatin structure.

### 3.3. Non-Coding RNAs

ncRNAs are generated from intergenic regions, repetitive sequences, TEs, and pseudogenes, and account for >90% of all RNA transcripts. As the name implies, ncRNAs do not encode proteins but exert regulatory effects in diverse biological processes. Thousands of ncRNAs have been identified and characterized in many plant species, including Arabidopsis, rice, soybean (*G**lycine max*), and maize [107]. ncRNAs are classified based on their length into short ncRNAs (sRNAs, 18–30 nt), medium ncRNAs (31–200 nt), and long ncRNAs (lncRNAs, >200 nt). The major types of sRNAs in plants include microRNAs (miRNAs) and TE-derived siRNAs, which differ in their biogenesis and modes of action [107]. miRNAs and lncRNAs downregulate target genes involved in ABA-mediated responses, osmoprotection, and antioxidant defense in response to drought stress [108,109].

TE-derived siRNAs participate in the transcriptional silencing of TEs by RdDM [110,111]. Plant TEs contain stress-responsive *cis*-acting elements in promoter regions and produce ncRNAs in response to stress [107,109,111]. Transcripts from TEs are processed into sRNAs, such as miRNAs and siRNAs. In rice, the TE-derived miR820 targets de novo DNA methyltransferase OsDRM2 transcripts, which are components of epigenetic silencing downregulated by stress [110]. Sharma et al. [112] showed that the overexpression of miR820 enhanced salt tolerance in rice plants. Zhang et al. [113] reported that TE-derived siRNA815 induced de novo DNA methylation via the RdDM pathway.

miRNAs interact with their targets to inhibit gene expression at the transcriptional, posttranscriptional, and epigenetic levels by inducing the cleavage of mRNA or repressing translation [11]. Genome-wide studies identified numerous up- or downregulated miRNAs in response to drought stress in Arabidopsis [114,115]. For instance, miR168 and miR396 contain ABRE *cis*-elements in promoter regions and are upregulated by drought stress [115]. Arabidopsis miRNA159 mediates the cleavage of *AtMYB101* and *AtMYB33* transcripts encoding MYB transcription factors that positively regulate the ABA response [116]. Li et al. [117] showed that miR169 targets the gene encoding the transcription factor NFYA5 (nuclear factor Y-A5), thereby repressing its transcription. miR169 was downregulated and NFYA transcript was strongly upregulated by drought stress in an ABA-dependent manner, thus enhancing the expression of several drought-responsive genes. In rice, many miRNA gene families are significantly up- or downregulated under drought stress [118,119,120,121]. For instance, Liu et al. [122] identified several miRNAs conserved in Arabidopsis and rice and reported that miR162 and miR167 were downregulated, whereas miR413 was upregulated, by ABA in rice. Xia et al. [123] showed that rice OsmiR393 targets two rice auxin receptor genes (*OsTIR1* and *OsAFB2*), which may decrease tolerance to drought and salt in addition to auxin sensitivity, thereby increasing tillers and early flowering. Yue et al. [124] showed that the inhibition or knockout of OsmiR535 in rice enhances tolerance to dehydration, ABA, and salt stresses. In addition, Yu et al. [125] reported that the overexpression of soybean miR169c increases the sensitivity to the drought stress of transgenic Arabidopsis.

Plant lncRNAs are transcribed by the plant-specific Pol II, Pol IV, and Pol V, leading to transcriptional gene silencing [126]. lncRNAs regulate alternative splicing of their target RNAs, affect chromatin topology, and modulate the transcription of neighboring genes [127,128]. lncRNAs regulate gene expression in the cytoplasm and nucleus via various mechanisms [11,126]. Cytoplasmic lncRNAs inhibit protein translation or act as miRNA mimics to inhibit miRNA activity and promote target mRNA translation. Nuclear lncRNAs modulate gene expression by affecting chromatin remodeling, and through epigenetic modifications and alternative splicing. In addition, lncRNAs are precursors of miRNAs and other sRNAs. lncRNAs are classified into three groups according to their positions relative to protein-encoding genes. Long intergenic ncRNAs (lincRNAs) have no overlap with the target genes, long intronic ncRNAs are synthesized from intronic regions, and long non-coding natural antisense transcripts (lncNATs) are synthesized from the opposite strand of the associated genes [128]. lncNATs bind to complementary mRNAs to trigger their degradation or promote their translation via the recruitment of polysomes [129,130].

Numerous lncRNAs responsive to drought, salinity, heat, cold, and ABA stresses have been identified in plants [107,131,132]. Qin et al. [133] identified the DRIR (drought-induced lncRNA) in Arabidopsis, which is localized mainly in the nucleus. An activation mutant (*drir^D^*) and DRIR-overexpressing lines were more sensitive to ABA and had a lower transpirational water loss rate than the wild type, and exhibited improved tolerance to drought and salt stress. Under salt stress, these lines showed altered expression levels of genes involved in ABA signaling, water transport, and the stomatal response to ABA. Wibowo et al. [67] found that certain regions of the Arabidopsis genome are susceptible to DNA (de)methylation in response to hyperosmotic stress. Using the *CNI1/ATL31* locus as an example, they demonstrated that epigenetically targeted sequences function as distantly acting control elements of antisense lncRNAs, which in turn regulate target gene expression in response to stress.

### 3.4. Chromatin Loop Formation

Regulatory elements in distal genomic regions can form a loop by physically interacting with the chromatin fiber, thus regulating gene expression [134,135,136]. Chromatin interactions are classified into two main types based on the distance between two regions: short- and long-range loops. Both types of chromatin interactions occur in Arabidopsis [137].

Short-range chromatin loops are formed by interactions in a single locus. This type of chromatin loop is formed by the interaction of the 5′- and 3′-flanking regions (promoter–terminator interaction) of a gene, thus facilitating the return of RNA polymerase from the terminator to the promoter region [138,139,140]. Tan-Wong et al. [141] observed promoter–3′-end chromatin loops in two yeast genes, which dynamically altered their expression. Short-range chromatin loops can be disrupted by environmental or endogenous signals, thereby promoting or suppressing gene transcription. Crevillén et al. [142] identified a gene loop involving a physical interaction of the 5′- and 3′-flanking regions of the Arabidopsis floral repressor gene (*FLC*) locus, which is disrupted during vernalization, resulting in the epigenetic silencing of the gene. Jégu et al. [143] showed that BAF60, a subunit of the SWI/SNF chromatin-remodeling complex, suppresses chromatin looping at the *FLC* locus.

Long-range chromatin interactions can occur within or between chromosomes. Typically, a chromatin loop provides distant regulatory sequences (e.g., enhancer, silencer, or regular protein binding sites) to the counterpart promoters, thereby enhancing or suppressing gene transcription [134,144]. This type of chromatin loop is formed by the binding of proteins (e.g., transcription factors) to distant regulatory sequences and gene promoters [145,146]. PcG complexes modulate transcription factor binding or chromatin regulatory loops in certain genome regions. In Arabidopsis, Guo et al. [147] reported that the transcription factor AG (AGAMOUS) and the PcG protein LHP1 bind to two specific regions flanking the *WUS* (*WUSCHEL*) gene body to form a chromatin loop. AG recruits and physically interacts with LHP1, and LHP1 binding to the chromatin loop is dependent on AG. Local chromatin looping at the *WUS* chromatin prevented the recruitment of RNA polymerase, thereby suppressing the gene transcription. Ariel et al. [148] showed that the lincRNA *APOLO* (*auxin-regulated promoter loop*) is transcribed by Pol II and V in response to auxin, a phytohormone controlling plant development, and regulates the formation of a chromatin loop encompassing the promoter of its neighboring gene *PID* (*PINOID*).

## 4. Transcriptional Memory of Drought Tolerance

### 4.1. Drought Stress Memory

Plants memorize the tolerance induced by drought stress to respond more effectively to subsequent stresses [13,149,150]. For instance, Walter et al. [151] observed that after a late drought during the growth of perennial grass (*Arrhenatherum elatius*), the percentage of living biomass was increased in plants exposed to earlier drought compared to those without such exposure, even after harvest and resprouting after the first drought. Ding et al. [152] reported that Arabidopsis with experience of dehydration stress wilted more slowly than plants without such experience in response to subsequent dehydration events. Wang et al. [153] showed that wheat plants subjected to one or two drought episodes before anthesis had higher grain yields under drought conditions. Ramírez et al. [154] reported that long-term stress improved drought tolerance-related traits and tuber yield in later growth stages in potato plants. Abdallah et al. [155] showed that the pre-exposure to a drought-sensitive variety of olive plants to drought enhanced their tolerance to subsequent drought conditions, resulting in improvements in biomass production, photosynthesis, and the maintenance of water status. Tabassum et al. [156] also reported that terminal drought and seed priming improved the drought tolerance of wheat plants.

Goh et al. [157] observed that Arabidopsis exhibited memory functions related to repeated ABA stresses, i.e., the impairment of light-induced stomatal opening and the induction of the expression of drought-responsive genes. Virlouvet and Fromm [158] reported that Arabidopsis stomatal apertures closed following exposure to dehydration remained partially closed during a recovery period with access to water, thereby facilitating reduced transpiration during subsequent dehydration stress. In mutant plants defective in the ABA signaling pathway, the guard cell stomatal memory was ABA-dependent, and *SnRK2*s were essential for implementing stress memory during the subsequent dehydration response. Li et al. [159] performed whole-transcriptome, strand-specific RNA sequencing (ssRNA-seq) of the rice genome, and the results suggested that lncRNAs, DNA methylation, and endogenous ABA mediate drought memory by activating the drought-responsive transcription of genes in pathways such as photosynthesis and proline biosynthesis in response to subsequent drought conditions.

### 4.2. Drought Stress Memory Genes

In response to drought stress, plants induce or suppress the expression of many drought-responsive genes [1,2]. In most cases, up- or downregulated gene transcripts return to basal levels during recovery (watered) states. However, a subset of genes is expressed at highly elevated or reduced levels in response to repeated drought stresses, which enables the plant to respond more promptly and strongly [13,16,152]. Ding et al. [160] defined stress memory genes as those that enable responses during subsequent stress conditions that differ from the responses during the initial stress encounter, whereas genes that respond similarly to each stress are categorized as non-memory genes. However, the expression threshold for classification into one or the other category is unclear.

Numerous drought stress memory genes have been identified in plants. Ding et al. [160] used a genome-wide RNA sequencing (RNA-seq) approach to evaluate the transcriptional responses of Arabidopsis leaves detached from plants repeatedly exposed to air-drying. Genes implicated in the responses to ABA, drought, salinity, and cold/heat acclimation constituted the drought-induced memory genes, and those responsible for chloroplast and thylakoid membrane-associated functions comprised the dehydration-repressed memory genes. Kim et al. [161] performed a microarray analysis to screen drought stress memory genes in soybean. The soybean memory genes exhibiting significantly elevated transcript levels upon the second exposure to drought stress conditions include those involved in ABA-mediated tolerance responses to abiotic stresses, such as genes encoding transcription factors, trehalose biosynthesis enzymes, late embryogenesis abundant proteins, and PP2C family proteins. By contrast, memory genes with highly reduced transcript levels during the second drought included genes involved in photosynthesis and primary metabolism. Soybean drought stress memory genes included genes involved in the dehydration memory responses of Arabidopsis. However, studies of other crop plants identified species-specific drought memory genes. A genome-wide RNA-seq analysis of maize identified only 4 chloroplast- and 2 thylakoid membrane-localized genes acting as drought-repressed memory genes [162], compared to 128 Arabidopsis drought-repressed memory genes [160]. In potato, the expression levels of most photosynthesis-related genes during a second drought were higher than during the first drought [163]. In addition, most rice memory transcripts associated with photosynthesis were markedly reduced by a first drought but then recovered, remaining at a stable level during subsequent drought treatments [159].

Several memory genes encode various transcription factors in all of the plant species described above. In Arabidopsis, dehydration-induced transcriptional memory behavior was seen in members of the AP2/ERF, bHLH, homeo_ZIP, MYB, ZF, b_ZIP, CCAAT, and WRKY transcription factor families [160]. Similarly, in soybean, various transcription factor genes belonging to the AP2, NAM, MYB, bZIP_1, and WRKY families were identified as drought-induced memory genes [161]. Therefore, transcription factors with memory function may contribute to plant memory and regulate the expression of their targets upon repeated stress. This possibility should be addressed in further studies of the mechanisms of drought stress memory.

### 4.3. Mechanism of Transcriptional Stress Memory

#### 4.3.1. Epigenetic Marks for Stress Memory

Epigenetic chromatin remodeling is a plausible molecular mechanism of transcriptional stress memory [12,22,23,150]. It has been proposed that chromatin architecture (DNA methylation, histone modification, and chromatin loops) maintains the altered gene expression patterns caused by an initial stressor. Transcription factors and chromatin loops may also be associated with a subset of their targets during mitosis. The role of the chromatin remodeler BRM in stress memory under heat-shock stress has been investigated. Brzezinka et al. [164] reported that Arabidopsis *brm* mutants were deficient in heat-shock memory and showed the reduced induction of heat-shock memory genes. BRM and the FORGETTER1 (FGT1) factor physically interact and are pre-associated with memory genes under non-stress conditions. *fgt1* mutants displayed more rapid recovery of nucleosome occupancy at heat-shock memory gene loci, suggesting that the BRM-FGT1 interaction prevents nucleosome recovery at these loci and mediates heat stress-induced memory.

Changes in DNA methylation may be involved in the transcriptional memory of plant responses to abiotic stresses [72]. Wang et al. [63] observed that around 29% of drought-induced DNA (de)methylation sites remained after recovery from drought or salt stress, implying that the DNA methylation changes were recorded. Kou et al. [165] performed a genome-wide rice methylome profiling analysis under recurrent drought stresses and recovery treatments. Most drought-stress memory-related DMRs were targeted TEs and few were targeted gene bodies, which suggests that they regulate TE expression to cope with recurrent drought stress. The distances from memory DMRs to TEs were significantly shorter than those from non-memory DMRs, implicating DNA methylation in drought memory formation.

Histone modification may provide a persistent epigenetic transmission mechanism associated with transcriptional memory in response to osmotic stress [18,20,21]. For instance, H3K27me3 is a gene silencing mark related to the chromatin-induced repression of gene expression and the formation of an epigenetic memory system during development [166]. In eukaryotes, the H3K27 methylation level is regulated by the action of polycomb group (PcG) protein complexes. Polycomb-repressive complex 2 (PRC2) mediates the deposition of H3K27me2/3 by the enzymatic subunit PRC2-Ezh2 (enhancer of ZESTE 2), whereas PRC2-Ezh1 restores H3K27me2/3 via its demethylase activity or histone exchange [167]. Ezh1 and Ezh2 exhibit different expression patterns and distinct chromatin-binding properties [168]. H3K27me3 in target genes recruits an additional PcG protein complex, PRC1. PRC1 complexes are subdivided into canonical PRC1 and noncanonical (or variant) PRC1. Canonical PRC1 is recruited by H3K27me3 readers and compacts nucleosomes to repress gene expression, while noncanonical PRC1 is recruited to chromatin independently of PRC2 and H3K27me3, and ubiquitylates histone H2A (to form H2AK119ub) via its H2A E3 ubiquitin ligase activity [169,170]. Thus, PcG complexes provide the major chromatin regulatory mechanism for silencing unnecessary or unwanted gene expression in mammals and plants [171,172].

PcG genes were discovered in *Drosophila* (*Drosophila melanogaster*), and homologs of PcG components and their target genes have been identified in other eukaryotes including plants [173,174,175,176]. The role of the PRC2-mediated deposition of H3K27me3 has been studied in the context of developmental processes and environmental stress responses in plant model species including Arabidopsis [172,177,178,179]. The PRC1-like protein LHP1 (like heterochromatin protein-1) was also identified in Arabidopsis [166,180,181]. Ramirez-Prado et al. [182] showed that the loss of *LHP1* induces ABA sensitivity and drought tolerance, indicating that LHP1 regulates the expression of stress-responsive genes. The H3K27me3 level was not related to H3K4me3 accumulation, suggesting that these histone modification marks function independently and do not have mutual effects on the expression of dehydration stress memory genes.

H3K4me3 deposition may play a role in the epigenetic transmission of active transcriptional states [19]. H3K4me3 deposition and the amount of Pol II stalled in memory genes were higher than in non-memory genes in Arabidopsis after multiple dehydration stresses [152,183,184]. Kim et al. [185] observed H3K4me3 and H3K9ac deposition in drought-inducible genes (*RD20*, *RD29A*, and *AtGOLS2*) in response to drought. During recovery by rehydration, H3K9ac was rapidly removed, whereas H3K4me3 was maintained at a low level, implying that H3K4me3 functions as an epigenetic mark of stress. Ding et al. [152] observed that H3K4me3 deposition in trainable genes (*RD29B* and *RAB18*) was maintained during recovery from stress-induced transcription but decreased to a basal level in non-trainable genes (*RD29A* and *COR15A*). Thus, H3K4me3 and Pol II were induced in several dehydration stress memory genes in response to the first dehydration stress event, persisted during the recovery period, and increased greatly due to a second stress event.

As identified in *Drosophila*, the epigenetic transmission of active transcriptional states is supposed to be mediated by TrxG (trithorax group) complexes: the SWI/SNF complex and the COMPASS (complex of proteins associated with Set1) family [186,187]. Antagonistic links were identified between PcG genes and SWI/SNF, and COMPASS was associated with histone methyltransferase activity leading to the H3K4me3 deposition. In Arabidopsis, multiple TrxG factors have been identified, based on their ability to suppress PcG mutant phenotypes [188]. Plant TrxG factors regulate gene transcription in seedling growth, anther and ovule formation, gametophyte development, and reprogramming during developmental transitions [189,190,191,192].

However, it is not clear whether H3K4me3 in chromatin contributes to transcriptional activation under subsequent stress conditions. Genome-wide transcript profiling revealed that the transcription of most genes is unaffected by the loss of the histone methyltransferase activity of ATX1 (Arabidopsis homolog of TRITHORAX 1) and that H3K4me3 is required for efficient elongation of the transcription, but not the initiation, of ATX1-regulated genes [193,194]. Howe et al. [195] proposed that H3K4me3 deposition in chromatin is a consequence of transcription, influencing splicing, transcription termination, the memory of previous states, and transcriptional consistency, rather than inducing gene transcription in response to repeated stresses. Moreover, the mechanism of TrxG recruitment to the chromatin during mitosis is not clearly elucidated.

#### 4.3.2. Memory Transmission by Mitosis

The stress-tolerance memory of a mother cell can be transmitted to daughter cells by mitosis. During mitosis, most transcriptional activities are diminished, and regulatory proteins are dissociated from chromosomes. However, the chromosomes in daughter cells are rearranged into cell type-specific conformations. Epigenetic bookmarks of specific gene regulatory elements may be maintained during this process [24]. Such mitotic bookmarks include DNA methylation, histone modifications, histone variants, ncRNAs, and certain transcription factors that remain bound during mitosis [25]. Kundu et al. [196] found that the ATP-dependent chromatin remodeling enzyme, SWI/SNF, is essential for transcriptional memory in the yeast GAL1 gene cluster.

Bellec et al. [197] proposed a hypothetical mechanism by which transcriptional memory facilitates post-mitotic gene reactivation. A previously transcribed locus may remain accessible during mitosis, and that histone marks and chromatin regulators of a chromatin state bind mitotic chromatin, thus bookmarking loci for subsequent transcriptional activation or silencing. Petruk et al. [198] observed that TrxG and PcG proteins but not methylated histones remain associated with DNA through replication, but H3K4me3 or H3K27me3 present during transcription were replaced by unmethylated H3 following DNA replication in *Drosophila*. Later, Black et al. [199] showed that the methylated histones remained in close proximity to DNA in *Drosophila* embryos during all phases of mitosis. In plants, Costa and Dean [200] reported that the expression of Arabidopsis *FLC* encoding a transcriptional repressor that delays flowering was epigenetically silenced during vernalization through a mechanism requiring H3K27me3 and PcG complexes (PRC2) and that H3K27me3 deposition at this gene locus was transmitted via cell division. Sani et al. [201] showed that the hyperosmotic priming of Arabidopsis seedlings by transient mild salt treatment led to the shortening and fractionation of H3K27me3 islands, thereby enhancing tolerance to a second stress (drought). Liu et al. [183] reported that the H3K27me3 level in Arabidopsis memory genes, such as *RD29B* and *RAB18* (*responsive to ABA 18*), increased upon first exposure to drought stress but remained consistent during the recovery period and a second exposure. The high level of H3K27me3 in transcriptionally inactive states did not interfere with the transition to active transcription or prevent the transcription of dehydration stress-responding genes.

Chromatin loops formed post-transcriptionally in mitotic chromosomes are supposed to be maintained throughout cell division and can then fulfill a memory function by preserving the active or repressed state of gene expression. Tan-Wong et al. [141] reported that the chromatin loops of two yeast genes were maintained for up to 1 h during an intervening repression period. When the yeast cells were re-supplied with transcription-induction factors during this period, RNA polymerases were recruited more rapidly to reactivate gene transcription. Deng and Blobel [145] proposed potential modes whereby the long-range chromatin loops are transmitted through mitosis; (1) chromatin loop structure persists throughout the cell division, (2) chromatin loops dissolve during mitosis, but the epigenetic marks remain bound to chromatin to direct loop reformation, (3) chromatin loops dissolve with the removal of looping factors, and epigenetic bookmarks facilitate loop reassembly following cell division.

Reflecting the observations in invertebrates, mammals, and plant flowering, we suppose a hypothetical mechanism of mitotic transmission for transcriptional memory (Figure 3). In this model, epigenetic marks (e.g., H3K27me3 or H3K4me3) and chromatin regulators (e.g., PcG complexes or TrxG complexes) remain bound on the chromatin loops formed in response to stress and are transmitted through mitotic cell division. In addition, certain transcription factors and chromatin remodeling complexes remaining in close proximity could access the chromatin of a memory locus, establishing a subsequent transcriptional status (e.g., silencing or activation). However, there is little evidence to support the notion that chromatin loop formation contributes to the memory of drought stress tolerance in plants. Therefore, our speculative model should be verified in further studies of the mechanisms of drought stress memory.

## 5. Transgenerational Inheritance of Memory

### 5.1. Transgenerational Transmission of Drought Tolerance

Traits acquired under stressful conditions can be transmitted to plant progeny [29,30,202]. The progeny of parents exposed to stress exhibits a higher yield than the progeny of non-stressed parents. Lämke and Bäurle [21] use the term ‘intergenerational memory’ when only the first stress-free generation has a detectable memory effect. As the progeny develops on the mother plant, intergenerational memory may be mediated by the conditions in which the seed grows and by cues introduced into the seed or embryo by the mother plant. In ‘transgenerational memory’, by contrast, memory is detectable after at least two stress-free generations. Verkest et al. [203] improved drought tolerance in canola (*Brassica napus*) by repeatedly selecting lines exhibiting increased drought tolerance for three generations. Tabassum et al. [204] reported that the hydro- and osmo-priming of bread wheat seeds caused the transgenerational transmission of improved tolerance to drought and salt stresses. Raju et al. [205] applied RNAi suppression to modulate abiotic stress-response pathways in soybean and developed an epigenetic breeding system for increased yield and stability.

Transgenerational memory may have an epigenetic basis [21]. Based on invertebrate developmental processes, it has been suggested that histones and other core chromatin components survive the passage of replication forks during meiosis [206,207,208]. Zenk et al. [209] showed that H3K27me3 was transgenerationally inherited from the maternal germline and resisted reprogramming events, thereby regulating the activation of enhancers and lineage-specific genes during early embryogenesis in *Drosophila*. Molla-Herman et al. [210] proposed that chromatin modifiers and Piwi-interacting small RNAs (piRNAs) function in adaptive and inheritable epigenetic memory events that occur in *Drosophila* during embryogenesis. Weiser and Kim [211] revealed an important role of endogenous siRNAs and piRNAs in transgenerational epigenetic inheritance in *Caenorhabditis elegans*. Those sRNAs may regulate heritable chromatin marks conveying epigenetic memory and thereby repress deleterious transcripts, such as TEs and repetitive elements. However, the relevance of these mechanisms to plant transgenerational inheritance of drought stress tolerance is unclear.

### 5.2. DNA Methylation for Transgenerational Inheritance

DNA methylation may contribute to transgenerational memory in plants [72]. Zheng et al. [212] reported that a high proportion of multigenerational drought-induced alterations in DNA methylation status are maintained in subsequent generations, possibly improving drought adaptability in rice. Drought stress-induced non-random epimutations over 11 successive generations improved the drought adaptability of rice epimutation lines. A large proportion (~45%) of the altered DNA methylation states were transmitted to unstressed progeny in subsequent generations. The epimutated genes participated in stress-responsive pathways, suggesting that they promote progeny adaptation to drought stress. Mathieu et al. [213] observed that the Arabidopsis mutant *methyltransferase 1-3* (*met 1-3*), which is deficient in terms of maintaining CG methylation, formed progressively more aberrant epigenetic patterns over several generations, suggesting that CG methylation is a central coordinator of epigenetic memory that secures stable transgenerational inheritance. Zhang et al. [214] reported that many epigenetic recombinant inbred lines of Arabidopsis were nearly isogenic as a result of drought but were highly variable at the level of DNA methylation. Cortijo et al. [215] identified several DMRs that act as epigenetic quantitative trait loci and account for 60–90% of the heritability related to flowering time and primary root length in Arabidopsis. However, the contributions of locus-specific methylation changes to the maintenance of stress memory and whether the inheritance of drought stress tolerance is mediated only by DMRs require further investigation.

Matzke and Mosher [216] proposed that RdDM contributes to the transmission of DNA methylation patterns in parental cells to their offspring by affecting germ-cell specification and parent-specific gene expression. Morgado et al. [217] observed that the composition of sRNAs in apomictic dandelion (*Taraxacum officinale*) lineages indicated a footprint of drought stress experienced two generations prior. Kuhlmann et al. [218] reported that the methylation of the reporter gene ProNOS was not completely erased in *DRM-2* (*domains rearranged methyltransferase 2*) mutants but persisted in the context of symmetric CG. ProNOS DNA methylation maintenance was evident after two generations of ongoing RdDM and increased in subsequent generations. They suggested that the methylation of a particular genomic region can be consolidated by RdDM and maintained over generations in Arabidopsis, thereby establishing epigenetic transgenerational memory. Wibowo et al. [67] suggested that epigenetic inheritance relies on DNA methylation changes at sequences that function as distantly acting control elements of key stress-response regulators, including antisense lncRNAs. Some of these changes are associated with conditionally heritable adaptive phenotypic stress responses and transmitted to the offspring, where they affect the transcriptional regulation of a small group of genes associated with enhanced tolerance to environmental stresses.

By contrast, two studies of Arabidopsis yielded conflicting results. In Arabidopsis subjected to slow-onset water deprivation treatment, Ganguly et al. [219] observed far fewer conserved DMRs in drought-exposed lineages compared to non-exposed lineages. Most of the variation was attributed to preexisting differences in the epigenome at repetitive regions of the genome. Thus, transgenerational memory may not be associated with changes in the DNA methylome. Van Dooren et al. [61] found that descendants of stressed and non-stressed Arabidopsis plants were phenotypically indistinguishable after an intervening generation without stress, irrespective of whether they were grown under normal or water-deficit conditions. In addition, although mild drought induced changes in the DNA methylome of exposed plants, these were not inherited by the next generation. Therefore, whether stress-induced DNA methylation variation transmits drought stress memory to the next generation is unclear.

### 5.3. Overcoming Meiosis

In mammals, paternal chromatin is extensively reprogrammed via the global erasure of DNA methylation achieved through extensive DNA demethylation and the packaging by exchange of histones with protamines [220,221]; this hampers the inheritance of stress-induced changes in chromatin architecture. Thus, epigenetic marks are reprogrammed in the gametes and the genomic potential is thus reset in the next generation. In contrast to mammals, DNA methylation in flowering plants is not completely erased from the germlines and is thus maintained during reproduction [222,223,224]. Wibowo et al. [67] reported that hyperosmotic stress memory in Arabidopsis restricts DME activity in the male germline. Moreover, protamine exchange does not occur in Arabidopsis, enabling the retention of histone-based chromatin in sperm [225]. However, Borg et al. [226] found that H3K27me3 is completely lost from histone-based sperm chromatin in Arabidopsis by the concerted action of three mechanisms; (1) the loss of histone methyltransferase activity in PRC2 to write H3K27me3, (2) the erasing activity of H3K27 demethylases, and (3) the deposition of the sperm-specific histone variant H3.10 which may be resistant to H3K27 methylation. The loss of H3K27me3 facilitates the transcription of genes essential for spermatogenesis but resets epigenetic memory in plant paternal chromatin.

Based on the above observations, newly acquired stress tolerance and associated epigenetic marks might be preferentially transmitted through the female germline. In Arabidopsis, Borg et al. [226] detected H3K27me3 in the microspore and in the daughter nuclei following microspore division, suggesting inheritance via meiosis. Inoue et al. [227] identified maternal H3K27me3 as a DNA methylation-independent imprinting mechanism in mouse (*Mus musculus*) embryonic cell lineage. Grossniklaus and Paro [228] reported that no major loss of H3K27me3 is expected on maternal alleles because PRC2 is active in the central cell. PcG complexes deposit or bind to certain histone modifications (e.g., H3K27me3 and H2AK119ub1) to prevent gene activation and maintain the repression of chromatin domains, which are implicated in plant vernalization and seed development. The relevance to plant drought stress tolerance mechanisms identified in flowering warrants further investigation.

## 6. Summary and Perspectives

Plant cells transmit acquired stress-tolerance traits to newly developed cells and the next generation. This could be exploited to improve agronomic technology and crop breeding because epimutations can be artificially induced more rapidly than through traditional genetic manipulation. However, the underlying principles and mechanisms are unclear, which hampers practical application. The most plausible molecular mechanisms underlying memory formation and transmission are epigenetic changes in memory gene loci in response to the initial stress. Such epigenetic marks include chromatin remodeling (e.g., DNA methylation, histone modifications, histone variant exchange), ncRNA generation, and chromatin loop formation. As illustrated in invertebrates, mammals, and flowering plants, histone marks (e.g., H3K4me3, H3K27me3), and chromatin regulators (e.g., PcG complexes, TrxG complexes) may remain bound on the chromatin during mitotic and meiotic processes and thereby play a role in the epigenetic memory transmission. It appears that in flowering plants, in contrast to mammals, remodeled chromatin structure is not completely erased from the germlines and is thus maintained during reproduction. The information gained from the plant flowering process warrants further investigation of drought stress tolerance mechanisms. The elucidation of the molecular mechanisms underlying drought stress tolerance, memory, and inheritance would facilitate the breeding of crops that can withstand global climate change.

## Figures and Tables

**Figure 1 ijms-23-12918-f001:**
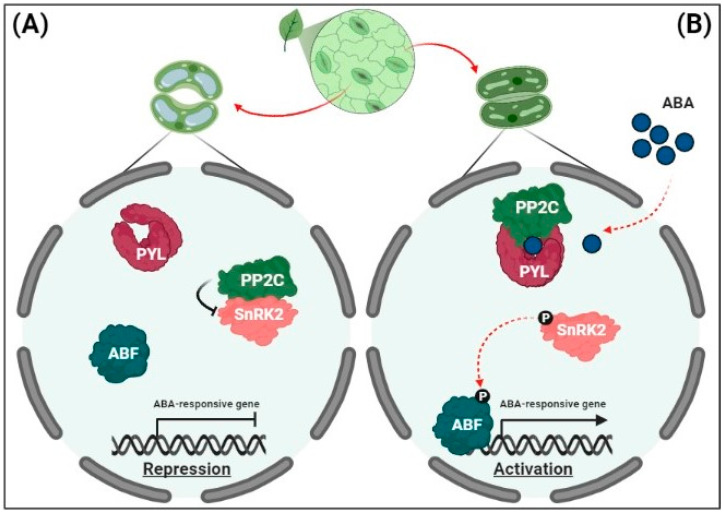
Signaling pathway for the expression of abscisic acid (ABA)-responsive genes. (**A**) Repression of ABA-responsive genes. In the absence of ABA, clade A protein phosphatases (PP2Cs) physically interact with sucrose non-fermenting 1-related protein kinase 2s (SnRK2s) to reduce kinase activity via dephosphorylation. This results in the inhibition of ABRE-binding (AREB)/ABRE-binding factors (ABFs) and the suppression of ABA-responsive gene transcription. (**B**) The activation of ABA-responsive gene expression. Under drought stress, soluble ABA receptors (Pyrabactin resistance/PYR-related/regulatory component of the ABA receptors [PYR/PYL/RCARs]) and PP2Cs act as co-receptors to capture ABA, thereby blocking the phosphatase activity of PP2Cs. PP2Cs are released from PP2C-SnRK2 complexes, and free SnRK2s phosphorylate downstream transcription factors (AREBs/ABFs). The phosphorylated AREBs/ABFs trigger the transcription of numerous ABA-responsive genes, leading to ABA responses including stomatal closure.

**Figure 2 ijms-23-12918-f002:**
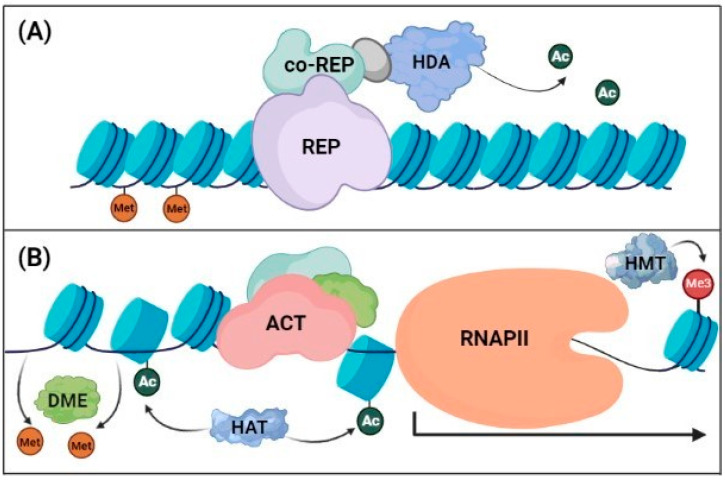
Hypothetical model of chromatin remodeling for the regulation of gene transcription. (**A**) Repression of gene transcription. Under normal conditions, promoter DNA is methylated, and repressors (REPs) or REP-corepressor (co-REP) complexes recruit histone deacetylases (HDAs) to compact the local chromatin region, leading to gene suppression. (**B**) Activation of gene transcription. For gene transcription, promoter DNA is demethylated by DNA glycosylase (DME). Repressors are released from the promoter, and activators (ACTs) or activator complexes bind to the open promoter region, which facilitates RNA polymerase II (RNAPII) access to initiate gene transcription. Histones are acetylated by histone acetyltransferases (HATs) to weaken DNA–histone interactions or are methylated by histone methyltransferases (HMTs).

**Figure 3 ijms-23-12918-f003:**
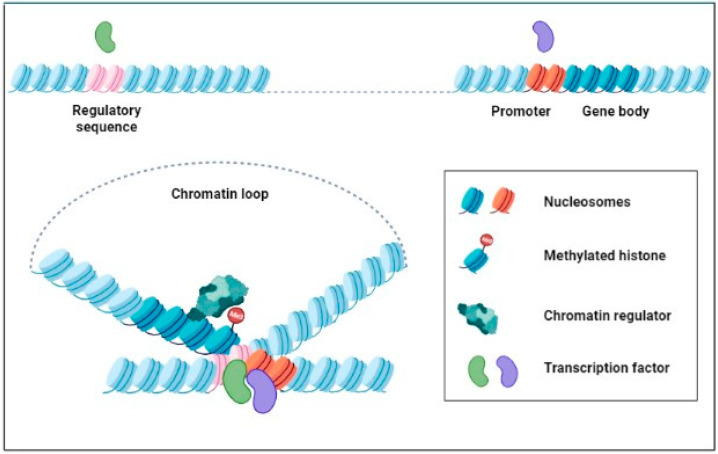
Hypothetical mechanism of mitotic transmission for transcriptional memory. In response to stress, long-range chromatin loops are formed by the interactions of multiple transcription factors that bind to distant regulatory sequences (e.g., enhancer, silencer, or regular protein binding sites) and the counterpart promoters, thereby enhancing or suppressing gene transcription. The chromatin loop can persist throughout the cell division cycle or dissolve during mitosis but reassemble following cell division. The chromatin of a memory locus may remain accessible to epigenetic marks (e.g., H3K27me3 or H3K4me3) and chromatin regulators (e.g., PcG complexes or TrxG complexes), thus bookmarking the locus for subsequent transcriptional status (e.g., silencing or activation). Histone octamers wrapped by DNA at the regulatory region (pink), promoter (orange), and gene body (deep blue) are represented, respectively, with different colors.

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
