# Peer review of "Transcriptional Stress Memory and Transgenerational Inheritance of Drought Tolerance in Plants"

_ijms, 2022, doi:10.3390/ijms232112918_

Round 1
Reviewer 1 Report
General comments:
This manuscript reviews updates knowledge regarding the role of epigenetic regulation of genes in response to drought stress. The MS is very well written and organized containing literature landmarks covering crucial epigenetics questions. The topics/titles addressed are really very well chosen. Complex concepts related to epigenetics marks and stress memory are clearly explained and supported on extensive data from the literature. The interesting perspective of future application of this knowledge for plant breeding purposes is also mentioned.
Only a few comments:
a. The term “memory genes” appears in the introduction and only later (page 10) is explained its definition.
b. The authors can develop a little more on the concept of “tolerance priming”.
c. Regarding the transgenerational inheritance, what is the threshold to classify as settled memory and for how many generations can the memory be maintained?
d. In figure 1, the letters A and B are mentioned in the legend but are missing in the figure. Also, in figure 1, check the spelling of “representation”
e. The tolerant genotypes exhibit not only hypomethylation but also higher ability for flexibility of DNA methylation levels.
f. the meaning of histone modifications in relation to gene transcription is also dependent on the specie. It’s difficult to establish a universal rule.
g. the authors could eventually elucidate a little more on the meaning of “long term stress memory”
Author Response
- The term “memory genes” appears in the introduction and only later (page 10) is explained its definition.
The sentence was supplemented to be “However, the most plausible mechanism involves epigenetic changes in the chromatin architecture of certain stress-responsive genes called ‘stress memory genes’ that are expressed at highly elevated or reduced levels in response to repeated stress [18–21].” [Page 2, 2nd paragraph]
- The authors can develop a little more on the concept of “tolerance priming”.
We change the sentence to “However, the adoption of tolerance priming (e.g., pre-treatment with stressful conditions) and inheritance technologies has been hampered by an insufficient understanding of their principles.” [Page 2, 4th paragraph]
- Regarding the transgenerational inheritance, what is the threshold to classify as settled memory and for how many generations can the memory be maintained?
The first paragraph of Section 5.1 [Page 15] was rewritten as below:
“Traits acquired under stressful conditions can be transmitted to plant progeny [29,30,202]. The progeny of parents exposed to stress exhibits a higher yield than the progeny of non-stressed parents. Lämke and Bäurle [21] use the term ‘intergenerational memory’ when only the first stress-free generation has a detectable memory effect. As the progeny develops on the mother plant, intergenerational memory may be mediated by the conditions in which the seed grows and by cues introduced into the seed or embryo by the mother plant. In ‘transgenerational memory’, by contrast, memory is detectable after at least two stress-free generations. Verkest et al. [203] improved drought tolerance in canola (Brassica napus) by repeatedly selecting lines exhibiting increased drought tolerance for three generations. Tabassum et al. [204] reported that hydro- and osmo-priming of bread wheat seeds caused the transgenerational transmission of improved tolerance to drought and salt stresses. Raju et al. [205] applied RNAi suppression to modulate abiotic stress-response pathways in soybean and developed an epigenetic breeding system for increased yield and stability.”
- In figure 1, the letters A and B are mentioned in the legend but are missing in the figure. Also, in figure 1, check the spelling of “representation”
The letters (A) and (B) are replaced by (Left side) and (Right side), respectively, in the legend of Figure 1.
- The tolerant genotypes exhibit not only hypomethylation but also higher ability for flexibility of DNA methylation levels.
The sentence suggested is inserted in the text, “The tolerant genotypes exhibit not only hypomethylation but also a higher ability for flexibility of DNA methylation levels.” [Page 5, 2nd paragraph]
- The meaning of histone modifications in relation to gene transcription is also dependent on the species. It’s difficult to establish a universal rule.
The first paragraph of Section 3.2.1 [Page 5] was rewritten as below: “In eukaryotic cells, genomic DNA is packed into the nucleus by wrapping around histone (H) octamers consisting of H2A, H2B, H3, and H4, thereby forming nucleosomes. To alter the activity of a gene, the N-terminal side of basic residues such as lysine and arginine in histone tails are covalently modified by acetylation, methylation, phosphorylation, and ubiquitination. Histone modification can have different effects depending on the type of modification and which residue is modified. It is difficult to establish a universal rule for the histone modification effects because the meaning of histone modifications in relation to gene transcription is dependent on the species.”
- The authors could eventually elucidate a little more on the meaning of “long term stress memory”
We delete the words ‘long term stress memory, and use ‘transcriptional stress memory’ in the revised manuscript:
“During and after stress, defense signaling metabolites and transcription factors accumulate in plant tissues, and may play a role in transient or short-term memory. However, the most plausible mechanism involves epigenetic changes in the chromatin architecture of certain stress-responsive genes called ‘stress memory genes’ that are expressed at highly elevated or reduced levels in response to repeated stress [18–21]. Lämke and Bäurle [21] defined the sustained differential responses in gene expression (activation or repression) after an exogenous cue with the term ‘transcriptional memory’.” [Page 2, 2nd paragraph]
“Epigenetic chromatin remodeling is a plausible molecular mechanism of transcriptional stress memory [12,22,23,150].” [Page 12, Section 4.3.1, 1st paragraph]

Reviewer 2 Report
Manuscript ijms-1918844,
“Epigenetic Regulation of Gene Transcription for Drought Tolerance in Plants”.
This is a thorough, well-written and well-organized review. It provides detailed information about the discoveries concerning transcriptional stress memory and transgenerational inheritance of drought tolerance in plants based on epigenetic mechanisms. This is an important and very interesting topic. Knowledge in the field of stress-related plant transcriptional memory could greatly contribute to crop management and improvement. The manuscript carefully reviews the studies related on the molecular and epigenetic basis of this phenomenon.
For the above reasons, the topic of this manuscript is important and relevant for IJMS, section Molecular Plant Sciences. I thus believe that the paper can be published after the authors address comments and edits from Reviewers.
Specific comments:
1) I noticed that there are at least three recent reviews on a very similar topic. I recommend that the authors explain what is unique in their review as compared to these three (in response to the reviewer). In addition, I think that it is necessary that the authors highlight and explain the novelty of this review in the Introduction section.
Yao et al. 2021 Transcriptional Regulation of Drought Response in Arabidopsis and Woody Plants https://www.frontiersin.org/articles/10.3389/fpls.2020.572137/full
Sun et al. 2021 Exploration of Epigenetics for Improvement of Drought and Other Stress Resistance in Crops: A Review https://www.ncbi.nlm.nih.gov/pmc/articles/PMC8235456/
Halder et al. 2022 Chromatin-Based Transcriptional Reprogramming in Plants under Abiotic Stresses https://www.mdpi.com/2223-7747/11/11/1449/htm
2) I recommend that the title should be changed to a more interesting and attractive for readers. I recommend something like “Transcriptional stress memory and transgenerational inheritance of drought tolerance in plants”.
3) Please briefly mention functions of aba-responsive genes and the mode of ABA membrane transportation in plant cells in the Section 2.1 of the manuscript.
4) Paper citations are needed for the following statements: “Non-CG methylation in CHG and CHH sites
occurs exclusively in heterochromatin.” (page 4)
5) Include citations at the end of this sentence: “High-resolution genome-wide analyses
suggest that methylation of gene promoter regions suppresses transcription of the corre-
sponding gene, whereas methylation of the gene body may enhance transcription.” (page 4)
6) Correct “To initiate the RdDM, a 24-nucleotide (nt) small interfering RNA (siRNA) is produced by RNA polymerase IV…” TO “To initiate the RdDM, 24-nucleotide (nt) small interfering RNAs (siRNA)s are produced by RNA polymerase IV…” (page 4)
7) Section 3.2.1. Histone modifications
Please briefly mention the known types of histone modifications for plants (at the beginning of this section)
8) Correct “Conversely, to initiate transcription of a gene, the surrounding chromatin needs to
be open to enable access by transcriptional activators and RNA polymerases” TO “Conversely, to initiate transcription of a gene, the surrounding chromatin needs to be openED to enable access by transcriptional activators and RNA polymerases”. (page 5)
It appears that “opened” is better to change – maybe “remodeled”?
9) Page 10, Correct “In response to drought-stress, plants induce…” TO “In response to drought stress, plants induce…”
10) Page 12, Correct ” However, whether H3K4me3 in chromatin contributes to transcriptional activation under subsequent stress conditions is unclear.” TO “However, it is not clear whether H3K4me3 in chromatin contributes to transcriptional activation under subsequent stress conditions.”
11) Page 14, Correct “Raju et al. [205] applied RNAi suppression to modulate abiotic stress-response pathways in soybean, and developed an epigenetic breeding system for increased yield and stability.” TO “Raju et al. [205] applied RNAi suppression to modulate abiotic stress-response pathways
in soybean and developed an epigenetic breeding system for increased yield and stability.” (comma deleted)
12) Page 16, Correct “Based on the above, newly acquired stress tolerance and associated epigenetic marks might be preferentially transmitted through the female germline.” TO “Based on the above knowledge(?), newly acquired ….”
13) Page 16, Correct “As illustrated in invertebrates, mammals, and plant flowering, histone marks (e.g., H3K4me3, H3K27me3) and….” TO “As illustrated in invertebrates, mammals, and flowering plants, histone marks (e.g., H3K4me3, H3K27me3) and…”
14) Page 16, Correct “The relevance to plant drought-stress tolerance mechanisms in plant flowering warrants further investigation.” Did you mean “plant flowering” or “flowering plants”? In addition the word “the relevance to” appears not appropriate here. Please rewrite this sentence.
15) I think that it would be great if the authors hypothesize why in plants the epigenetic marks are not erased during meiosis, while in mammals they are erased and reset during meiosis (in Summary section).
Author Response
1) I noticed that there are at least three recent reviews on a very similar topic. I recommend that the authors explain what is unique in their review as compared to these three (in response to the reviewer). In addition, I think that it is necessary that the authors highlight and explain the novelty of this review in the Introduction section.
The three review articles are cited in the Introduction and included in the Reference list.
[Page 2, the last paragraph of the Introduction part]
“Recently, Yao et al. [31] summarized recent progress in the regulation of drought response by various transcription factor families in plants. Halder et al. [32] reviewed recent studies on chromatin restructuring under abiotic stresses, crosstalk between epigenetic regulators, and stress priming in plants. In addition, Sun et al. [33] summarized and discussed mechanisms involved in plant stress memory and genome editing technologies with their future possible applications, in the breeding of crops for abiotic stress tolerance. Here we review the research on the epigenetic molecular mechanisms underlying transcriptional drought stress memory in plants. In particular, we infer a hypothetical concept for the possible mechanisms underlying the epigenetic regulation of transcriptional and transgenerational transmission of drought stress tolerance, using associations with observations of plant reproductive process and reprogramming events in germ cells of other animal systems.”
2) I recommend that the title should be changed to a more interesting and attractive for readers. I recommend something like “Transcriptional stress memory and transgenerational inheritance of drought tolerance in plants”.
The title has been changed as recommended.
3) Please briefly mention functions of aba-responsive genes and the mode of ABA membrane transportation in plant cells.
Functions of ABA-responsive genes and the mode of ABA membrane transportation are described in the revised manuscript.
[Page 2, Section 2.1, First paragraph]
“ABA induces the expression of numerous genes encoding enzymes that catalyze the biosynthesis of osmoprotectants such as trehalose and late embryogenesis abundant proteins. In addition, ABA-induced genes include those encoding proteins exerting either positive or negative effects on its accumulation (biosynthesis, catabolism, and glucose-conjugation), transportation, and signaling network [35].”
[Page 3, Section 2.2, First paragraph]
“ABA is biosynthesized in vascular tissues and transported to targets including guard cells via the export and import of transporters. In Arabidopsis, three transporter families have been identified: ATP binding cassette (ABC), detoxification efflux carriers (DTX)/multidrug and toxic compound extrusion (MATE) proteins, and nitrate transporter 1/peptide transporter family (NPF) [35, 41–431. For instance, ABCG25 exports ABA to the apoplastic space in vascular tissues, while ABCG40 and NPF4.6 import ABA into guard cells in response to osmotic stress.”
4) Paper citations are needed for the following statements: “Non-CG methylation in CHG and CHH sites
occurs exclusively in heterochromatin.” (page 4)
This document was found in the Reference [19] (page 7, first paragraph) where any reference was indicated. This sentence may not be necessary for the text, thus has been deleted in the revised manuscript. [Page 4, Section 3.1, 1st paragraph]
5) Include citations at the end of this sentence: “High-resolution genome-wide analyses
suggest that methylation of gene promoter regions suppresses transcription of the corresponding gene, whereas methylation of the gene body may enhance transcription.” (page 4)
We put the reference numbers [58–60] at the end of the sentence. [Page 4, the last line]
6) Correct “To initiate the RdDM, a 24-nucleotide (nt) small interfering RNA (siRNA) is produced by RNA polymerase IV…” TO “To initiate the RdDM, 24-nucleotide (nt) small interfering RNAs (siRNA)s are produced by RNA polymerase IV…” (page 4)
The sentence has been corrected as pointed out. [Page 4, Section 3.1, the first paragraph]
7) Section 3.2.1. Histone modifications. Please briefly mention the known types of histone modifications for plants (at the beginning of this section)
The first paragraph of the Section 3.2.1 has been re-written
“In eukaryotic cells, genomic DNA is packed into the nucleus by wrapping around histone (H) octamers consisting of H2A, H2B, H3, and H4, thereby forming nucleosomes. To alter the activity of a gene, the N-terminal side of basic residues such as lysine and arginine in histone tails are covalently modified by acetylation, methylation, phosphorylation, and ubiquitination. Histone modification can have different effects depending on the type of modification and which residue is modified. It is difficult to establish a universal rule for the histone modification effects because the meaning of histone modifications in relation to gene transcription is dependent on the species.”
8) Correct “Conversely, to initiate transcription of a gene, the surrounding chromatin needs to
be open to enable access by transcriptional activators and RNA polymerases” TO “Conversely, to initiate transcription of a gene, the surrounding chromatin needs to be opened to enable access by transcriptional activators and RNA polymerases”. (page 5) It appears that “opened” is better to change – maybe “remodeled”?
The word ‘opened’ has been changed to ‘remodeled] in the revised manuscript. [Page 6 the last paragraph]
9) Page 10, Correct “In response to drought-stress, plants induce…” TO “In response to drought stress, plants induce…”
The words ‘drought-stress’ have been changed to ‘drought stress’ in the revised manuscript.
10) Page 12, Correct ” However, whether H3K4me3 in chromatin contributes to transcriptional activation under subsequent stress conditions is unclear.” TO “However, it is not clear whether H3K4me3 in chromatin contributes to transcriptional activation under subsequent stress conditions.”
The sentence has been corrected as pointed out. [Page 13, 4th paragraph]
11) Page 14, Correct “Raju et al. [205] applied RNAi suppression to modulate abiotic stress-response pathways in soybean, and developed an epigenetic breeding system for increased yield and stability.” TO “Raju et al. [205] applied RNAi suppression to modulate abiotic stress-response pathways in soybean and developed an epigenetic breeding system for increased yield and stability.” (comma deleted)
The comma has been deleted in the revised manuscript. [Page 15, Section 5.1, the first paragraph]
12) Page 16, Correct “Based on the above, newly acquired stress tolerance and associated epigenetic marks might be preferentially transmitted through the female germline.” TO “Based on the above knowledge(?), newly acquired ….”
We put a word ‘observations’ after the word ‘above’ to be “Based on the above observations,---“ [Page 17, Section 5.3, 2nd paragraph]
13) Page 16, Correct “As illustrated in invertebrates, mammals, and plant flowering, histone marks (e.g., H3K4me3, H3K27me3) and….” TO “As illustrated in invertebrates, mammals, and flowering plants, histone marks (e.g., H3K4me3, H3K27me3) and…”
We changed the words ‘plant flowering’ into ‘flowering plants’, as pointed out. [Page 17, the last paragraph]
14) Page 16, Correct “The relevance to plant drought-stress tolerance mechanisms in plant flowering warrants further investigation.” Did you mean “plant flowering” or “flowering plants”? In addition, the word “the relevance to” appears not appropriate here. Please rewrite this sentence.
This sentence was rewritten to be “The information gained from the plant flowering process warrants further investigation of drought stress tolerance mechanisms.” [Page 17, the last paragraph]
15) I think that it would be great if the authors hypothesize why in plants the epigenetic marks are not erased during meiosis, while in mammals they are erased and reset during meiosis (in Summary section).
We add one sentence in the Summary part: “It appears that in flowering plants, in contrast to mammals, remodeled chromatin structure is not completely erased from the germlines and is thus maintained during reproduction.” [Page 17, the last paragraph]
